# Gender and Financialization of the Criminal Justice System

**Lisa Servon \*, Ava Esquier \* and Gillian Tiley**

Department of City and Regional Planning, University of Pennsylvania, Philadelphia, PA 19104, USA; gtiley@design.upenn.edu
\* Correspondence: servon@design.upenn.edu (L.S.); aesquier@sas.upenn.edu (A.E.)

**Abstract:** (1) The increase in women's mass incarceration over the past forty years raises questions about how justice-involved women experience the financial aspects of the criminal justice system. (2) We conducted in-depth interviews with twenty justice-involved women and seven criminal law and reentry professionals, and conducted courtroom observations in southeastern Pennsylvania. (3) The results from this exploratory research reveal that women's roles as caregivers, their greater health needs, and higher likelihood of being poor creates barriers to paying fines and fees and exacerbates challenges in reentry. (4) These challenges contribute to a cycle of prolonged justice involvement and financial instability.

**Keywords:** gender-specific; mass incarceration; reentry; fines and fees; poverty; criminal justice system





## 1. Introduction

Between 1978 and 2015, the number of women in the U.S. correctional population rose by 834 percent, more than twice the rate of men. Two million women and girls return home every year; one million women currently live under probation, parole, and correctional supervision (Kajstura 2019; The Sentencing Project 2019). During this same period, the criminal justice system became increasingly financialized, evidenced by the growth in private prisons and widespread use of legal financial obligations (LFOs) such as fines and fees (Friedman and Pattillo 2019; Wang 2018; Harris 2016).

Most research on incarceration and reentry focuses on men or on the justice-involved population as a whole and ignores the unique circumstances of women (Kajstura 2019; Swavola et al. 2016). This exploratory research begins to fill that gap. Specifically, we explore how women's reentry is complicated by their inferior economic status, their caregiving responsibilities, and the justice-related debt they have accumulated. The analysis of in-depth interviews with twenty justice-involved women, seven criminal law and reentry professionals, and courtroom observations in southeastern Pennsylvania reveals that fines, costs, and fees coupled with preexisting issues such as caregiving and trauma, create intractable reentry barriers for women, contributing to a cycle of prolonged justice involvement and financial instability. In this paper, we examine the implications of financial precarity throughout the stages of subjects' justice involvement, highlighting the unique experiences of justice-involved women and the role of the financialization of the criminal justice system in amplifying participants' challenges.

Although, since 1980, the growth rate of female imprisonment has been twice as high as that of men (The Sentencing Project 2019), research and policy debates typically focus on men, in part because the number of incarcerated men far exceeds the number of incarcerated women. The rapid increase in women's incarceration resulted from: (1) the gendered and racialized nature of the War on Drugs; and (2) expanded policing and intensified prosecution of minor crimes such as conspiracy. The War on Drugs and 'tough-on-crime' movement, initiated in the 1980s, triggered a decades-long period of mass incarceration in the United States that impacted both women and men. However,

policies driving the War on Drugs incriminated women at higher rates than men: between 1980 and 2009, drug-related crimes increased threefold for women and twofold for men (Merolla 2008; Moe and Ferraro 2006; Swavola et al. 2016; Wakefield and Uggen 2010). Mandatory minimum sentencing, the increased policing of minor crimes, and expanded minimal involvement charges (such as conspiracy and accomplice liability) have disproportionately affected women over time (National Resource Center on Justice Involved Women 2016; Swavola et al. 2016; Jacobs 2017; Cobbina 2009). In 2011, for example, 25.1% of women in state prisons were incarcerated for a drug offense, compared to 16.2% of men (Ramirez 2016).

Black and Latinx women were arrested at higher rates than white women as a result of these policies, despite similar rates of drug use across racial and ethnic groups (National Resource Center on Justice Involved Women 2016; Merolla 2008; Kajstura 2019). This historical pattern of over-policing and over-incarcerating Black and Latinx women is verified (Haley 2016). These groups are overrepresented in correctional populations: Black women are imprisoned at twice the rate of white women, and Latinx women at 1.3 times the rate of white women. At the same time, the imprisonment rate for Black women declined since 2000, while it increased for white and Latinx women (The Sentencing Project 2019). In proportion to the overall population, indigenous women are incarcerated at six times the rate of white women (Hartney and Vuong 2009).

Compared to men, women's offenses during this period were more likely to be nonviolent. According to national data from 1980 to 2008, women comprised just 11 percent of violent offenders (Cooper and Smith 2011). Increased economic disadvantages among women are correlated with higher rates of property crime (Steffensmeier 1993; Steffensmeier and Haynie 2000). Furthermore, women were more likely to be charged with "partner" crimes such as conspiracy and economic survival crimes such as stealing food and clothing (Allen et al. 2010; Swavola et al. 2016). Studies show that women commit "survival crimes" to provide for their families and escape abuse, and/or to feed drug addiction, which often stems from abuse (Belknap 2020; DeVuono-Powell et al. 2015; Richie et al. 2000).

Moreover, the feminization and criminalization of poverty provide important context for women's justice involvement and the impact of financialization (Cobbina 2009; Mallik-Kane and Visher 2008; Moe and Ferraro 2006; Pearce 1978). These factors correlate with trends in mass incarceration: among women in jail, the majority are poor, single caretakers, and women of color (Heimer 2000; Moe and Ferraro 2006; Spjeldnes et al. 2014). The feminization of poverty[1] refers to the disproportionate number of women in poverty (Pearce 1978; McLanahan and Kelly 1999). From 1980 to 2013, the number of female-headed households grew and nearly one in four unmarried mothers with children lived in poverty (Bleiweis et al. 2020). Racialized mass incarceration disproportionately affects Black and Latinx women who are more likely to be poor. Black women are five times more likely to live in poverty compared to white women, and about half of all Black and Latinx women have zero net wealth (Chang 2010).

Bloom and Covington (2008) suggest that women are increasingly implicated by the criminalization of poverty based on "survival" tactics related to trauma and poverty activities (Bloom and Covington 2008). For justice-involved women, crimes involving property, prostitution, and selling and using illicit drugs are often responses to partner violence, victimization, trauma, and economic insecurity (Belknap 2020; Daly 1996; Heimer 2000; Pollock 1998; Reckdenwald and Parker 2008; Scroggins and Malley 2010; Steffensmeier and Streifel 1992). However, there is no consensus among researchers regarding whether women are more likely to commit crimes of economic survival compared to men, despite experiencing higher poverty rates.

Throughout the period of mass incarceration, the criminal justice system underwent a period of financialization characterized by the increased levying of fines and fees, the privatization of prisons, and changes in the ways imprisoned people procured goods and services. Financialization shifted the costs of criminal justice functions onto defendants,

creating long-term debt, increasing justice involvement, and disproportionately burdening low-income communities of color (Harris et al. 2010).

During the 'tough-on-crime' era of the 1980s and 1990s, state and local governments instituted new laws codifying and intensifying what Alexes Harris calls "monetary sanctions"—legal financial obligations (LFOs), or fines and fees for justice involvement (Bannon and Diller 2010; Harris 2016). Monetary sanctions include fines for wrongdoing and restitution to reimburse victims for costs, fees charged to defendants in the criminal justice system, and the cost of prison phone calls (Harris 2016). While most criminal defendants are considered indigent, Harris et al. estimate that 66 percent of incarcerated individuals and 84 percent of those on probation owe fines and fees (Harris et al. 2010). Harris and others highlight the extreme variance in state law and county-level practices governing the imposition, enforcement, and collection practices of fines and fees by the courts, law enforcement, and public officials (Cadigan and Kirk 2020; Friedman and Pattillo 2019; Harris 2016; Martin et al. 2018; Pleggenkuhle 2018). Since 2008, 48 states have added or increased the costs of fines and fees for criminal and civil justice involvement (Menendez et al. 2019).

The privatization of justice system functions is a second way that financialization has become embedded in the criminal justice system (Harris et al. 2019). Defendants must pay bail bondsmen and private defense attorneys pre-incarceration. Inmates pay steep charges for commissary items, such as food and toiletries, phone calls while imprisoned, and probation services upon release. Private firms profit from these practices: inmates and their loved ones pay commissary vendors USD 1.6 billion annually (Wagner and Rabuy 2017).

Monetary sanctions create obstacles to successful reentry and trigger long-term debt, incarceration, and instability in reentry, especially for low-income defendants (Cadigan and Kirk 2020; Cook 2014; Harris 2016; Pleggenkuhle 2018). Indigent defendants are routinely assessed thousands of dollars in mandatory fines and fees by judges, without regard to their ability to pay (Bannon and Diller 2010; Harris et al. 2010; Harris 2016; ACLU-PA 2021). Nationally, the family of an incarcerated loved one pays, on average, over USD 13,000 in court-related costs and one in three go into debt to cover these costs (DeVuono-Powell et al. 2015). The consequences for nonpayment vary greatly by jurisdiction and even by judge, and include mandatory court appearances, driver's license suspension, additional late fees, probation extension, arrest warrants for missed court dates, and incarceration (Bannon and Diller 2010; Cook 2014; Friedman and Pattillo 2019; Harris 2016).

Harris (2016), Wang (2018) and others argue that the system of fines and fees is extractive and rooted in racial capitalism, burdening the Black and low-income communities that are disproportionately affected by mass incarceration (Harris 2016; Wang 2018). Wang (2018) points to the 2015 U.S. Department of Justice's Investigation of the Ferguson Police Department, which revealed a history of routine racial bias by public officials and police officers who targeted Black communities to generate revenue from fines and fees (United States Department of Justice Civil Rights Division 2015). The Ferguson case is not alone; the 2017 U.S. Commission on Civil Rights found that municipalities with higher shares of African American and Latino populations rely more heavily on revenue from fees and fines (US Commission on Civil Rights 2017).

Finally, the personal histories of justice-involved women differ from justice-involved men before, during, and after incarceration. It is well documented in the literature that women's economic and social marginalization prior to imprisonment exacerbates their challenges in reentry (Allen 2018; Severance 2004). Women lag behind men in education and job skills, making it more difficult to attain long-term, well-paid employment (Ramirez 2016; Wright et al. 2012). Women demonstrate greater needs than men in every area of reentry, particularly in health due to pregnancy, substance abuse dependency, and physical and mental health issues (Spjeldnes et al. 2014). Research suggests that women have more trouble than men finding affordable and safe housing due to childcare responsibilities, histories of abusive relationships, and drug-related histories (DeVuono-Powell et al. 2015; Freudenberg 2006). According to Freudenberg, childcare re-

sponsibilities " . . . make women more dependent on abusive male partners, [and] less able to work . . . and focus on their own health" (Freudenberg 2006, p. 14). Fewer halfway programs, shelters, mental health and substance abuse treatment centers are available for women as they leave prison, adding to the marginalization and stigma that justice-involved women face (Cowan n.d.; Kruttschnitt 2010; Ramirez 2016).

Although women are less likely to recidivate than men, financial instability and a lack of social capital among justice-involved women are linked to recidivism (Holtfreter et al. 2004; Johnson 2014; Piper Deschenes et al. 2006). One-third of women return to jail or prison within one year, and two-thirds recidivate within five years (Durose et al. 2014). Some researchers suggest that, compared to men, more women lose contact with family during incarceration and must rely on themselves to establish a home and gain stability when released (Cobbina and Bender 2012; Severance 2004). Other researchers attribute recidivism to gender differences in crime (such as consumer-based crimes), feminization of poverty, and drug dependence (Daly 1996; Holtfreter et al. 2004; Reisig et al. 2002).

## 2. Methods

### 2.1. Interviews

The core of our work consisted of twenty semi-structured, in-depth interviews, conducted with justice-involved women in the Philadelphia area and seven professionals with expertise in the areas of gender and incarceration. The interviews took place between March and June 2020. Due to COVID-19 travel restrictions, all interviews were conducted by telephone.

We recruited participants through staff and email listservs at various reentry organizations in Philadelphia, including the Philadelphia Reentry Coalition, Why Not Prosper, the People's Paper Co-op, and Sisters Returning Home. The interview protocol for the formerly incarcerated women focused on the financial challenges faced before, during, and after incarceration, including pre- and post-release debt, access to financial services, and types of financial support. In addition, we interviewed seven professionals in the fields of criminal law and reentry. Interviews lasted between 35 and 60 min. We conducted all interviews as "guided conversations", following up on participants' responses to questions (Rubin and Rubin 2012).

We recorded all of the interviews and had them professionally transcribed. Immediately following each interview, the interviewer compiled field notes to summarize the details of the interview. Using the grounded-theory method[2], we analyzed data in a two-step coding process using the coding software Dedoose (Glaser and Strauss 1967). At least two researchers coded each interview. Each participant was compensated with a digital USD 50 gift card.

### 2.2. Court Observations

Prior to COVID-19, we observed court hearings in Philadelphia and Lebanon County, Pennsylvania to understand how fines and fees are imposed and enforced by individual judges. The field sites were Lebanon County Courthouse, Philadelphia Traffic Court, and the Center for Criminal Justice in Philadelphia. At each courtroom session, we took fieldnotes and transcribed them afterwards.

We completed approximately 12 h of courtroom observations before we had to suspend these activities due to COVID-19 restrictions. Given the limited time, we were unable to generalize findings to themes across courtrooms. However, we can point to a major difference in court processing between Philadelphia and Lebanon County. Once a month, the Lebanon County Court of Common Pleas holds "Fines and Costs Contempt" hearings for those with unpaid debt. In Philadelphia, there are no court hearings devoted specifically to fines and costs. Instead, we attended bail hearings and traffic courts to investigate how judges set bail and traffic court payment plans. This study was approved by the University of Pennsylvania Institutional Review Board.

*2.3. Limitations*

This exploratory study is small, which limited our ability to generalize from the data and findings; it was conducted in a relatively small area of Pennsylvania. Differences in the ways the law is implemented from place to place made it impossible for us to draw broader conclusions. Our work was further hampered by the dearth of criminal justice data, as broken down by gender. However, the fact that the incarceration of women is growing nationally, and that financialization has broadly impacted the criminal justice system, we believe our findings are relevant and help to articulate further areas of research.

**3. Results**

*3.1. Crimes of Survival*

The majority of women we interviewed reported financial instability before their first incarceration. Fourteen out of twenty were employed or received income prior to their incarceration, but only eight considered themselves to be financially stable. Prior to justice involvement women are more likely to be primary caregivers of children, contributing to higher expenses and more difficulty finding work that accommodates their needs (Glaze and Maruschak 2010). Sixteen of the women we interviewed were financially responsible for children before their first incarceration. Courtney, a single mother of two, began engaging in prostitution as a means of survival for herself and her daughter:

> "So I was a single mother and just trying to take care of me and my daughter. It was really hard to do. It's almost impossible to pay all your bills . . . Eventually, I turned to other ways like prostitution to help me. That's what landed me in jail."

Monica, a single mother, was first arrested for stealing clothing for her children at a time when she could not afford basic needs for her family:

> "When I went to jail, my first charge, with going to jail for not even $50 worth of clothes. Even the cop who arrested me, he felt bad. And I didn't even make it out the store. I gave the things back. And he was like, 'You're just stealing stuff for your kids. It's really hard out here'."

Alyssa relayed a similar experience:

> "A lot of women I know that did prostitution, it wasn't to get high, it was to get money so they can take care of their family and they may have gotten arrested or people that shoplift, it's a way to make money, shoplift and sell things so you can have an income so you can take care of your family."

The crimes of economic necessity that participants committed often triggered long-term contact with the criminal justice system.

*3.2. Trauma and Addiction*

Incarcerated women are more likely than men to be poor and commit crimes related to life histories of poverty, victimization, mental health issues, and substance abuse (Anderson et al. 2020; Bloom and Covington 2008; Richie 2001; Wright et al. 2012). The interviews we conducted with both criminal justice professionals and justice-involved women illustrate the intersection between trauma-coping mechanisms and justice involvement. An estimated 99 percent of incarcerated women have experienced a traumatic event in their lives and 90 percent have experienced violence (Saxena et al. 2016). Reverend Dr. Michelle Simmons, director of Why Not Prosper, a community-based reentry services organization, spoke of her own path from trauma to incarceration: "I had suffered trauma. I had suffered physical and mental abuse, which led me to addiction and incarceration".

Just over half of the participants experienced drug addiction or substance abuse at some point in their lives. Uggen and Thompson (2003) and Daly (1996) found that women coped with pain and stress by using drugs, which often led to economic needs and criminal activity to sustain drug habits (Daly 1996; Uggen and Thompson 2003). Clarissa began using drugs to cope with financial hardship. Discussing her path to incarceration, she said: "There is a financial problem [because] really you're addicted to drugs. You're trying to raise

children [and] you're losing back and forth." The harsher drug and sentencing laws that were part of the War on Drugs accelerated the increase in women's incarceration because they punished their coping mechanisms without treating the root causes of the problem.

*3.3. Self-Reliance, Higher Expenses, and Financial Instability in Reentry*

Involvement with the criminal justice system appeared to negatively affect women's financial stability upon reentry. Twelve women reported being "financially stable" prior to incarceration. In reentry, 17 out of 20 participants described themselves as "financially unstable" or "in between" financial stability and instability. More than half of the women we interviewed received some financial support from family or friends post-incarceration. Still, 17 out of 20 described difficulties obtaining housing, caring for children, paying off court debt, and accessing basic services, especially for mental health and substance abuse treatment.

Women in reentry often manage family reunification and roles as single caretakers (Belknap 2020), while attempting to find employment and housing with little or no government assistance. Seven participants were responsible for children at the time of the interview. Anika, who spent multiple years in prison, described returning home as a single caretaker:

> "I'm doing this so much more by myself, I don't have the support of people, like the outside help. I didn't really have outside help when I was inside. I can't really call up and ask anybody for money to help me pay my bills ... I was left by myself with six kids and I didn't know what to do. I had no support. I had no inside support, I had no outside support."

The majority of children with incarcerated fathers live with their mothers; those whose mothers are imprisoned typically live with other family members or in foster care (Martin 2017). Mothers who are incarcerated, and who do not have custody of their children following incarceration, often owe child support, adding to their financial instability. Data from the National Conference of State Legislature suggest that over 400,000 incarcerated parents currently have a child support case ("Child Support and Incarceration" 2019). Elizabeth, a mother of two suffering from PTSD, talked about how her justice involvement affected her relationships with her children:

> "I'm just trying to elevate my mind beyond what they robbed me from, and to see my spirit. Because I just feel trapped. And my kids need me and I feel like can't do shit for them ... I'm a provider and a protector, and they took that from me. Took that from my children."

*3.4. Behavioral and Mental Health Needs throughout the Reentry Process*

Women have greater health needs than men in reentry, particularly for mental health issues and substance abuse treatment; these needs add to their expenses and compromise their ability to work. Spjeldnes et al. described the growing population of justice-involved women as more "medically and emotionally fragile", requiring gender-specific reentry services to reduce the risk of recidivism (Spjeldnes et al. 2014, p. 91). Amara described how her needs for income and services sometimes conflicted:

> "I'm transitioning from a life of addiction and I had no resumé ... It was like I was being born again. I had to do an outpatient program that I was stipulated to for 18 months when I came home. So I couldn't work right away, so it was difficult."

Amara attended a reentry program that focused on connecting women in an outpatient program with employment resources, and said that the program allotted too little time for healing:

> "It seemed like the only thing, a whole bunch of doctors and people who have never been in prison, they got together with this thought about what women need when they come home without asking people who have already been through that process ... There has to be a period where they can get their minds right. If

that period right there can be so stressful . . . financially, [people] can't make their ends meet. It causes people to go back to what they know to do to make money."

### 3.5. Employment

Half of the participants were unemployed and received no income or public assistance at the time of the interview. The majority of those who had found employment said their income was inadequate to meet expenses. Seven received income from government assistance programs such as Social Security and Supplemental Security Income (SSI). Only three participants had stable employment at the time of the interview. Several of the women discussed how difficult it was to find work that would enable them to support themselves and their families. Monica works in home care and discussed how traditional employment opportunities for women do not pay enough:

" . . . when men come home, . . . they could be a felon, and they can go get a construction job. They can go to the roofing job making great money, awesome money. Us women, we can't do that."

Jennifer described how she patches income together from a variety of sources:

"So, learning how to be financially responsible with the little bit of money . . . doing these kind of things . . . I do focus groups. I do a smoking study to help make a little bit of money, where I can put it either towards savings or things that I need, because not working, you still have to survive. You still have responsibilities."

Accessing limited job opportunities and holding a well-paid job is more difficult for those with childcare or child support responsibilities who have higher expenses and greater time obligations. Job opportunities are significantly limited for justice-involved women with children, as is accessing cash assistance on a minimum-wage income. Discussing the income limits for childcare support, Mary said:

"It's like you can get all kinds of daycare, but if you want to go out and get a job and feel proud of yourself and become self-sufficient, they cut off once you make . . . if you make $11 an hour, you're cut off of welfare."

When we asked participants what policy changes would increase their opportunities to succeed, well-paid jobs was the most common response.

### 3.6. Housing

Finding stable income affects and overlaps with struggles to find affordable and safe housing. The stigma of criminal conviction and the lack of social capital severely limits formerly incarcerated people's options and is especially challenging for women with children and histories of addiction. Peggy Sims, the founder of the reentry organization, Sisters Returning Home in Philadelphia, said her biggest challenge is finding housing for the women that her organization serves:

" . . . we're looking at this population like they're not homeless . . . They are homeless, because a lot of the families have just had enough. I don't want you stealing from me anymore, I don't want to talk with you anymore."

Sims added that housing is more difficult for women. According to her, the women she works with must rely on themselves upon first returning home, while for men: "There's always some woman waiting to take him home, dress him up . . . feed him, you know, and take care of him till they get a job." Reverend Dr. Michelle Simmons concurred that finding housing is often a gender-specific struggle:

" . . . men can come home and go anywhere. Women have to come home and get themselves settled so they can have a place for their children, so their mindset is a little different . . . women got to stand on their own, and men ain't really available to support the women, to embrace with their children."

Approximately half of the participants in our study experienced homelessness in reentry and lived or are currently living in a halfway house, recovery house, or shelter in reentry. In these homes, they may be restricted from employment, travel, and other freedoms, and may have to pay rent or give up their government assistance as a form of payment. Some women who have nowhere to go upon release extend their jail time because of the shelter and food they receive. Jo, who was incarcerated six times, described the role of jails housing the homeless:

> "What I noticed . . . was that a lot of women spend a lot of extra time in jail because their home plans weren't being accepted, or they just didn't have anywhere to go. They were homeless. So, if you don't have an address that you can go to, they'll hold you. They might just max you out."

The additional time spent in jail because of a lack of financial resources further disrupts women's lives.

Women's ability to regain custody of their children is also dependent on their finding a stable place to live, a difficult task for someone with a criminal record. Miranda experienced addiction and homelessness. Her outpatient obligations interfered with finding stable employment, while her childcare obligations constrained her permanent housing options:

> "Even if I apply for housing, it's a long wait . . . I suffer from drug addiction, so it might be an area where I don't want to raise my children and it might be in [an] area where drugs are heavy, so I might not be able to get housing. I might not be able to move to that area . . . It might be hard for me to get childcare."

Sophia explained the circular nature of the problem:

> " . . . that's emotional stress right there. Them thinking about their children and if they don't get a place to stay, they're not going to get their children back. But then it's like, how can you get a place to stay when jobs won't hire you with a criminal history?"

Our research supports Richie's notion of "the co-occurrence of multiple demands," in which returning women's urgent needs for childcare, housing, and healthcare "consume their material and emotional resources" (Richie 2001, p. 381). Women lack the support systems that men traditionally return to, while burdened by childcare responsibilities and family separation or reunification.

### 3.7. Criminal Justice Debt and Bail

Women's greater likelihood of poverty before incarceration, combined with their caregiving responsibilities and greater health needs, make it more difficult for them to pay fines and fees post-incarceration. Their diminished financial stability, triggered in part by the financial costs of justice involvement, made reentry more difficult. The inability to pay court-ordered costs created a cycle of financial struggle and prolonged criminal justice contact.

Justice-involved women are more likely to have lower incomes than justice-involved men. However, this hardship is often ignored by judges. The women we interviewed reported owing USD 300 to USD 129,000 in legal fines, fees, costs, restitution, and child support payments. The median amount owed was USD 4000. Fewer than half of the participants said that the court assessed their ability to pay, yet 15 out of 20 women were punished for nonpayment. In Pennsylvania, a judge must determine whether a defendant is willfully not paying before punishing them for unpaid fines and fees.[3] The lack of clear standards allow judges, who often ignore life circumstances, complete discretion in determining the ability to pay and punishing defendants who are unable to pay (ACLU-PA 2021; Friedman and Pattillo 2019). In our courtroom observations at fines and costs hearings in Lebanon County, Pennsylvania, judges asked defendants with outstanding payments about their employment status and life circumstances to determine why they could not pay. In some instances, defendants brought paperwork to substantiate their ability to pay and the judge

neither took nor looked at it. The judges we observed determined defendants' payment plans and punishments, sending one defendant to jail immediately for nonpayment.

In some jurisdictions, it is not the infraction itself that puts a woman behind bars but rather her inability to afford the required fines, fees, and cash bail. Mary, a 43-year-old mother, was fined USD 155 for her initial infraction, a drinking violation. She was jailed for nonpayment several times and ultimately owed USD 5000 in backlogged fines and fees. Discussing how her fees were determined, she said:

"[The amount of your debt] has nothing to do with your expenses. It has nothing to do with the amount of children you have."

Before becoming justice involved, Clarissa received income from government assistance and living in public housing, but was still unable to afford gas and utility bills. She explained how she used drugs to cope and, soon after, her children went to live with her family members. She was jailed for one year before her trial because she could not pay the USD 10,000 in cash bail. While incarcerated, she was evicted from her home and lost all of her belongings. She said:

" . . . a mom don't have enough for her baby, to wear deodorant because she had to pay gas and it's expensive. She had to pay a $200 gas bill or she got a shut off notice. She can't let her children be without hot water or food. I just feel like it should be looked at especially of people that's having hardship."

Our research revealed that the cost of court-ordered payments triggered long periods of incarceration and instability for the women we interviewed. Ella, a mother and recovering addict, explained the challenge of attaining financial stability following incarceration:

"So you get back on your feet and make enough money because in the beginning, you're definitely going to be living paycheck to paycheck, because a lot of times you starting all over again. You starting from scratch. So you need so much. You just can't go out and start a banking account . . . Then the job opportunities for ex-offenders aren't that lovely as it is. So you're not making it. You have to save up and build up or work more than one job. There's just so much . . . "

She continued, describing the lack of time allotted between being released from prison and beginning repayment: "I was trying to explain to the guy in traffic court, I just came home. Can you give me a minute to get on my feet and find a job before I start paying? . . . They want you to start paying within the next month."

All but two participants in this research were on court-mandated payment plans and owed between USD 5 and USD 50 per month. Fourteen women experienced stress about paying back their fines, costs, and fees, and only seven participants received financial assistance from family, friends, and community groups. About half of the women reported that paying back their court debt interfered with their ability to afford basic necessities. Those owing restitution or child support experienced additional stress and harsher consequences for not keeping up with their payment plans. Renae explained how the payment plan affected her ability to afford childcare:

"They [the payment plans] interfere greatly. I can't really do a lot of things. I do things a dollar at a time. Like I'm trying to move or if I'm trying to get a car or anything for me and my daughter, I have to almost save up for like a year to let that money build to something. Because maybe after I pay my expenses and pay my restitution, I might just have $20 left and I have to save it and let it build."

The consequences of nonpayment of fines and fees include work-release sentences, extended probation, bench warrants, garnished wages, denial of government assistance, re-arrest, re-imprisonment, and additional fees and fines. Clarissa owed USD 7000 in fines and fees upon her release. Because of her inability to pay what she owed, her probation was extended, and she was threatened with partial incarceration through work release. Similarly, Mary was incarcerated for not paying a drinking violation. She described a cycle of unpaid court debt and incarceration. After serving a maximum sentence for the charge, Mary had to pay USD 200 per month to be released from probation:

"You shouldn't go to jail that many times on that charge. And it was a drinking charge. And so, then my fines just got bigger and bigger, and a year probation turned into five years. And now, they're locking me up all the time they see me, and I don't have any money now. And now this $155 turned into $5000 . . . And then finally, eventually they said, listen. If you sit in jail for eight months, your fines will be paid . . . Eventually, it was over in five years."

## 4. Discussion

The women we interviewed expressed a shared sense of financial and legal precarity related to their justice involvement. We cannot separate the effects of the financialization of the criminal justice system from the impact of women's already precarious financial positions. Our research raises questions about the interaction between poverty, criminalization, and financialization and how it may exacerbate the destabilization of families and communities by prolonging the period of justice involvement and indebtedness. The precarity created by these three factors appears to stimulate a vicious cycle that contributes to and maintains the risk of incarceration. For the women we interviewed, the financialization of the criminal justice system appears to amplify the pre-existing financial precarity of justice involved women.

The issues created by the financialization of the criminal justice system aggravated women's situations at all stages of their justice involvement. Pre-incarceration, the women we interviewed faced the expenses of cash bail and court fees, which were often unattainable due to their existing financial instability. During incarceration, women faced the cost of commissary and phone calls, often having to divulge much of their income towards such costs or go without. Post-incarceration, women were burdened with fines and fees of the criminal justice system, assigned without an assessment of their ability to pay. With fines ranging from USD 300 to more than USD 100,000, women were either unable to pay or contributed much of their income to such expenses.

The financial struggles of justice-involved women are distinguished by (1) the personal histories of women in the justice system; (2) and the financialization of the criminal justice system. Compared to men, women face higher rates of victimization, substance abuse, mental and medical health issues, unemployment, and single parenthood. These issues arose in nearly every interview we conducted. Although we did not interview men for this study, our findings align with the literature demonstrating that the gendered experiences of women contribute to the cyclical justice involvement of women.

Women, whose financial precarity increases with justice involvement, often return to criminal activity after their release out of necessity. The personal histories of women amplify this cycle. Struggles with addiction, a lack of resources, and challenges in finding the housing and employment necessary to care for their families all work to further entrench women in financial instability. Low-income women engage in criminal activity to cover basic expenses. They become incarcerated, a process that adds debt, stigma, and other barriers to their pre-incarceration financial instability. The lack of reentry services and treatment coupled with increased financial precarity contributes to recidivism. Figure 1, below, illustrates this cycle.

**Figure 1.** The cycle of financial instability and justice involvement for women based on interview data.

## 5. Conclusions

This research begins to connect the dots between the financialization of the criminal justice system and a worsening of the already precarious financial and health situations of justice-involved women. Our results support Wesley and Dewey's findings, in which "arrest, incarceration, and subsequent acquisition of a criminal record compound rather than disrupt the vulnerabilities women face prior to their criminal justice system involvement" (Wesely and Dewey 2018, p. 62). The women we interviewed described how financial instability, family responsibilities, and limited social networks exacerbated challenges to meet mental and behavioral health, income, and housing needs. Their financial instability was also amplified by the debts owed to the increasingly financialized criminal justice system.

This exploratory research enabled us to identify areas for further research. First, some jurisdictions have begun to reduce or eliminate the debt-inducing elements of their criminal justice systems. These efforts should be studied to understand the long-term effects. Second, we hypothesize that women's greater challenges before incarceration cause them to be more impacted by the financial aspects of the criminal justice system than men, potentially contributing to a higher rate of recidivism. Comparative research that studies both women and men should be conducted to test this hypothesis. Finally, many of the women we interviewed for this study committed crimes of economic survival. There is no large-scale study that explores the commonness of such crimes and whether women commit them at a higher rate than men.

**Author Contributions:** Conceptualization, A.E. and L.S.; methodology, L.S.; formal analysis, A.E., L.S. and G.T.; investigation, A.E. and L.S.; data curation, A.E.; writing—original draft preparation, A.E. and L.S.; writing—A.E., L.S. and G.T.; visualization, A.E. and G.T.; supervision, L.S.; project administration, L.S.; funding acquisition, L.S. and A.E. All authors have read and agreed to the published version of the manuscript.

**Funding:** This research was funded by the Institute for Urban Research (IUR) at the Weitzman School of Design, University of Pennsylvania.

**Institutional Review Board Statement:** The study protocol was approved by the Institutional Review Board of the University of Pennsylvania.

**Informed Consent Statement:** Informed consent was obtained from interviews participants before participating in the study. Pseudonyms are used to protect the anonymity of the participants.

**Conflicts of Interest:** The authors declare no conflict of interest. The funders had no role in the design of the study; in the collection, analyses, or interpretation of data; in the writing of the manuscript, or in the decision to publish the results.

## Notes

[1]   Most refer to "feminization" as the gender poverty gap. By focusing on sex disparities by gross income and household headship, a largely U.S. definition of poverty, the term "feminization of poverty" is too narrow a definition to understand the multidimensionality of the global gender poverty gap. (McLanahan and Kelly 1999; Chant 2012; West et al. 2017).

[2]   In sociology, grounded theory in qualitative research is a method of generating concepts from data collection and comparative data analysis.

[3]   Many jurisdictions use the term "willful" when determining the consequences for nonpayment of fees and fines. Friedman and Pattillo (2019) argue that this term "both assumes an autonomous individual who is in full control of their circumstances and fixes the blame on the individual who acts with clear purpose" (p. 191).

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
