# Peer review of "Gender and Financialization of the Criminal Justice System"

_socsci, doi:10.3390/socsci10110446_

Round 1

Reviewer 1 Report

The manuscript is framed around some important questions about how the financialization of the criminal justice system is experienced (differentially) by women. The paper appropriately uses interviews (20) as a source of exploratory evidence.

This strategy draws out some interesting shared experiences of many of the women in this sample, which are described in section 3. It does not enable evaluating the hypothesis "that the financialization of the criminal justice system impacts justice involved women differently than men" (page 1). It lacks the comparison to men as well as to less financialized interactions with the CJS. If this hypothesis is emerging from the interview analysis rather than being tested by this analysis, then it would make more sense to put it in the discussion and to frame this as a testable implication for future work. But right now it's not really a hypothesis that is evaluated. This also makes the first sentence of the discussion (section 4) rather misleading, since the analysis doesn't really speak to a relationship between CJS financialization and recidivism.

On a related note, it's unclear from the analysis what role is being played by the financialization of the criminal justice system, versus the pre existing financial insecurity of the women, and I think that the author could do more to tease out these dimensions and foreground those concerns in the set up and discussion of the results.  Figure 1 is not very intuitive and also seems to suggest that financial instability, rather than financialization itself, is at the root of repeated criminal justice involvement. 

While the topic is important enough to warrant publishing even exploratory findings I worry that the author is claiming more than the data and analysis (at least as presented in this version) allow.

Author Response

Response to Reviewer 1 Comments

Point 1: This strategy draws out some interesting shared experiences of many of the women in this sample, which are described in section 3. It does not enable evaluating the hypothesis "that the financialization of the criminal justice system impacts justice involved women differently than men" (page 1). It lacks the comparison to men as well as to less financialized interactions with the CJS. If this hypothesis is emerging from the interview analysis rather than being tested by this analysis, then it would make more sense to put it in the discussion and to frame this as a testable implication for future work. But right now it's not really a hypothesis that is evaluated.

Response 1: We agree with this comment and have revised the language, replacing the previous hypothesis with the following: “In this paper, we examine the implications of financial precarity throughout the stages of subjects’ justice-involvement, highlighting the unique experiences of justice-involved women and the role of the financialization of the criminal justice system in amplifying participants’ challenges.”  We also added several recommendations for further research in the conclusion.

Point 2: This also makes the first sentence of the discussion (section 4) rather misleading, since the analysis doesn't really speak to a relationship between CJS financialization and recidivism.

Response 2: We agree and have revised the language here.  We also include research on the relationship between financialization and recidivism as an area for future research.

Point 3: On a related note, it's unclear from the analysis what role is being played by the financialization of the criminal justice system, versus the pre existing financial insecurity of the women, and I think that the author could do more to tease out these dimensions and foreground those concerns in the set up and discussion of the results.  

Response 3: Thank you for this suggestion. We have reworked the language in the paper to say that our findings begin to connect the dots between financialization and women’s already precarious financial situations.  We cannot separate the two factors in this exploratory work but have added more clear language in the set up and discussion.

Point 4: Figure 1 is not very intuitive and also seems to suggest that financial instability, rather than financialization itself, is at the root of repeated criminal justice involvement. 

Response 4: We agree and have revised Figure 1 to reflect our findings poverty and trauma-coping are often the root of repeated justice-involvement.

Point 5: While the topic is important enough to warrant publishing even exploratory findings I worry that the author is claiming more than the data and analysis (at least as presented in this version) allow.

Response 5: Thank you for this suggestion. We have changed the language to be more in line with the exploratory nature of the research, and are now careful not to claim more than the data allow.

Reviewer 2 Report

Please consider some main points to revise your paper to improve the quality and academic knowledge before publishing official the article. 

Page 3-4: Need to re-format to integrate with the whole of paper. I assumed that these are references' source (1-7) rather then their findings. If so, please re-reference as publishers' guidelines. 

Reference: Please check throughout your in-text citation (e.g., E.K. Allen 2018 on the 4th page). Please re-reference as publishers' guidelines. 

For ethics: I assumed that interview participants in judge-involved women and the authors compensated with a digital $50 gift card, etc., are part of the ethical approval before conducting this research. Therefore, it should be confirmed in the whole of your method's section rather than providing at the Court Observation's section (line221-222 on the sixth page).

For discussion and conclusion: while at the abstract, the authors stated: Results from this exploratory research reveal that women's roles as caregivers, more significant health needs, and the higher likelihood of being poor create barriers to paying fines and fees and exacerbates challenges in reentry.

The authors should be extended further details to compare and contrast with previous studies relating to the same topic to demonstrate their timely contribution via this case study in Pens. Around the half-page length of these two crucial sections (discussion and conclusion) are irrelevant and unbalancing with their scientific additions in the field. 

Author Response

Response to Reviewer 2 Comments

Point 1: Page 3-4: Need to re-format to integrate with the whole of paper. I assumed that these are references' source (1-7) rather then their findings. If so, please re-reference as publishers' guidelines. 

Response 1: We agree with this comment, and we have incorporated and re-referenced per the publishers’ guidelines.

Point 2: Reference: Please check throughout your in-text citation (e.g., E.K. Allen 2018 on the 4th page). Please re-reference as publishers' guidelines. 

Response 2: We agree with this comment, and we will incorporate and re-reference per the publishers’ guidelines.

Point 3: For ethics: I assumed that interview participants in judge-involved women and the authors compensated with a digital $50 gift card, etc., are part of the ethical approval before conducting this research. Therefore, it should be confirmed in the whole of your method's section rather than providing at the Court Observation's section (line221-222 on the sixth page).

Response 3: We agree with this comment. This information is now included in the methods section.

Point 4: For discussion and conclusion: while at the abstract, the authors stated: Results from this exploratory research reveal that women's roles as caregivers, more significant health needs, and the higher likelihood of being poor create barriers to paying fines and fees and exacerbates challenges in reentry. The authors should be extended further details to compare and contrast with previous studies relating to the same topic to demonstrate their timely contribution via this case study in Pens. Around the half-page length of these two crucial sections (discussion and conclusion) are irrelevant and unbalancing with their scientific additions in the field. 

Response 4: Thank you for this suggestion. We agree with this suggestion and want to point out the existing areas where our paper compares and contrasts with previous studies and proposes additional arguments. For example, we state on page 8, lines 383-385 that our results support Beth Richie’s argument based on evidence from her interviews with formerly incarcerated women. Given the severely understudied nature of this topic and population, we do not agree that extending compare and contrast with previous studies would strengthen this paper.

In response to your feedback, we would find it useful to include a sentence that this study confirms previous studies in that 1) justice-involved women experience greater challenges than justice-involved men, and 2) most people who have to pay fines and fees struggle to pay fines and fees. We now included a statement in the discussion  section on page 10, lines 493-495: “Although we did not interview men for this study, our findings align with literature demonstrating that the gendered experiences of women contribute to the cyclical justice involvement of women.” We also want to highlight the concluding statement on page 11 of the conclusion section, lines 511-514,which is responsive to your feedback: “Our results support Wesley and Dewey’s findings, in which “arrest, incarceration, and subsequent acquisition of a criminal record compound rather than disrupt the vulnerabilities women face prior to their criminal justice system involvement” (Wesely and Dewey 2018, 62).”

Round 2

Reviewer 1 Report

The author(s) did a satisfactory job of responding to my earlier comments. I have no further comments.